# LEAN IMAGES FOR GEO-LOCALIZATION

## ABSTRACT

Most computer vision tasks use textured images. In this paper we consider the geo-localization task – finding the pose of a camera in a large 3D scene from a single *lean image*, i.e. an image with no texture. We aim to experimentally explore whether texture and correlation between nearby images are necessary in a CNN-based solution for this task. Our results may give insight to the role of geometry (as opposed to textures) in a CNN-based geo-localization solution. Lean images are projections of a simple 3D model of a city. They contain solely information that relates to the geometry of the scene viewed (edges, faces, or relative depth). We find that the network is capable of estimating the camera pose from lean images for a relatively large number of locations (order of hundreds of thousands of images). The main contributions of this paper are: (i) demonstrating the power of CNNs for recovering camera pose using lean images; and (ii) providing insight into the role of geometry in the CNN learning process;

## 1 INTRODUCTION

Imagine you are brought blindfolded to a street corner of a city you know well. Now, you remove the blindfold. Can you tell where you are? This is the geo-localization task. In computer vision, this amounts to estimating the position (and sometimes the orientation) of a camera given its current view. Although localization devices such as Global Positioning Systems (GPS) have improved significantly over the last years, they often do not work well in city scenes and do not provide highly accurate results. Autonomous cars, drones, and IOT devices are expected to benefit tremendously from the ability to determine their pose (position & orientation) accurately in their environment.

A solution for geo-localization, either by a human or by a machine, can use appearance cues (*e.g.*, texture of a unique building), geometric cues (*e.g.*, a unique shape of a building), or both. Significant improvements were obtained in object recognition, scene recognition, and localization tasks, largely by exploiting the appearance of the scene rather than only its edges (*e.g.*, color and texture and image features such as SIFT Se et al. (2002); Lowe (2004); Li et al. (2010)). Later, these methods were improved by adding coarse geometric constraints to the image features (*e.g.*, Ramalingam et al. (2011); Bansal & Daniilidis (2014)). Nowadays, methods are based on machine learning, in particular convolutional neural networks (CNNs), where the input is typically the unprocessed textured image. Clearly, both appearance and geometry play an important role in these methods.

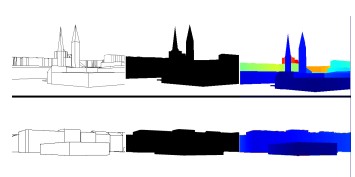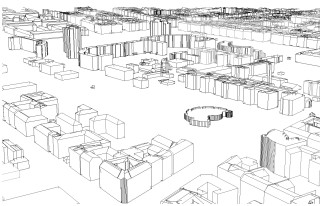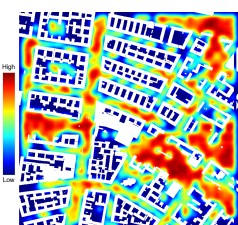

Figure 1: Left: *lean images* contain mostly geometric features: edges (left), faces (center), and depth information (right), which are projections of a 3D model. Center: Bird's-eye view of one of the areas we used. Right: a top view of a city area (buildings are marked as white) where color indicates the localization success rate of the network from red (high) to blue (low). Note how open spaces are more distinct than narrow streets.

The main goal of this paper is to study the role of geometry in a CNN solution to the geo-localization task rather than propose a working system for application purposes. We study an end-to-end deep CNN for geo-localization, by ignoring the often available texture of the scene. To do this, we consider only lean images, which contain mostly information that relates to the geometry of the scene while lacking texture or rich geometric details. In practice, we use a city scene and consider two types of binary images that consist of the edges of the buildings' outline and the buildings' facades. In addition, we also consider depth images that contain more geometric information. Examples of the three types of lean images are shown in Fig. 1 (left). Note that in the first row, the view contains dominant landmarks, while the second row shows very little distinct information that might be expected to assist localization. Such non-distinct views are very common in large environments such as a city, making localization with lean images very challenging. Further note that we deliberately do not consider real images or synthetic images with texture, since our goal is to study only the information available from pure geometric data.

Our lean-images are obtained simply by projecting an untextured 3D mesh model of Berlin onto various positions in the scene (See Sec. 3). A bird's-eye view of one of the areas is shown in Fig. 1 (middle). Using such a model allows us to study the role of geometry for geo-localization in a controlled manner and in a larger scale than ever before, both in terms of the area covered (many city streets) and in terms of the number of images (up to hundreds of thousands).

Neural networks were shown to be effective in geo-localization tasks (*e.g.*, Kendall et al. (2015); Walch et al. (2017); Melekhov et al. (2017); Sattler et al. (2017)). A typical geo-localization solution will return the pose of a camera in novel views. That is, the image query is not part of the training set. In this paper we ask (i) can the CNN generalize and support geo-localization in a large environments using lean images that contains only geometric and spatial data? (ii) is it likely that geometric information is used by the CNN to solve this task? However, to better understand the geo-localization task, we also consider the memorization capacity of the network for previously seen images. This can be regarded as image retrieval from a database of all available views of the city. A naïve solution would store all images and then perform a brute-force search in the database. However, this is inefficient and can become infeasible as the database gets larger. Defining a compact representation and an efficient search is clearly desired. The question we address in this case is (iii) whether CNNs can be used to solve the meomorization task using lean images.

As discussed in the results (Sec. 6), we found positive answers to all these questions, but the results depend on the number of images and their sampling density. We believe this indicates that networks can learn some sort of a spatial map for an area using only geometric data, since no colors or textures are available in our data. The success of geo-localization also depends on the specific position. Fig. 1 (right) shows how certain positions in the streets of a city are more recognizable than others.

Our paper presents an empirical study regarding the information that can be used by CNNs to build an internal representation; we do not propose a practical solution based on lean images. The main contributions of our study are: (i) showing that lean images contain sufficient information for solving the memorization and the geo-localization tasks (ii) proposing a systematic method to study the role of geometry in CNNs for geo-localization tasks; (iii) demonstrating the power of CNNs to use the geometric information to build internal represenations of data.

## 2 RELATED WORK

Place recognition (*e.g.*, the Eiffel Tower) can be regarded as a coarse geo-localization task.Classic approaches use visual features to represent each image in a set of images of a given location (*e.g.*, by a bag of words) and then match a target image with the stored representations (*e.g.*, Se et al. (2002); Sivic & Zisserman (2003); Lowe (2004); Robertson & Cipolla (2004)). Hays & Efros (2008) were the first to address the place recognition task using millions of geo-tagged images, based on various visual features (e.g., tiny images, color histograms, line features, gist descriptors).

Memorization of image data-sets is often performed by manually engineered image representation (*e.g.*, a dictionary of image features) and an image retrieval approach, including the metric between the stored representation and a target one. (*e.g.*, Sivic & Zisserman (2003); Lowe (2004); Robertson & Cipolla (2004); Zhang & Kosecka (2006); Nister & Stewenius (2006); Schindler et al. (2007);

Hays & Efros (2008); Li et al. (2010)). We show that neural networks are able to efficiently create such a representation in various experiments.

A possible solution for geo-localization task, where both position and orientation of a camera with respect to a scene should be estimated, is by using triangulation with images with known pose (*e.g.*, Zhang & Kosecka (2006)). In most studies, 3D models of the scene are used by means of point-clouds (*e.g.*, Irschara et al. (2009); Sattler et al. (2011); Matei et al. (2013); Svarm et al. (2014)), Digital Elevation Maps (DEM) (*e.g.*, Baatz et al. (2012); Bansal & Daniilidis (2014)), or full 3D models (*e.g.*, Ramalingam et al. (2011)). One of the main challenges of these works is to develop an efficient computation of 2D to 3D feature matching. New 3D feature representation have also been developed (*e.g.*, Irschara et al. (2009); Sattler et al. (2011)). Bansal & Daniilidis (2014) introduce a feature more closely related to the lean images we consider. It consists of 3D corners and direction vectors extracted from a Digital Elevation Map (DEM) to be matched geometrically to the corners and roof-line edges of buildings visible in a street-level query image. Efficiency and robustness become even more important when dealing with a city-scale 3D model. A fast method for inliers detection that enables solving the correspondence problem on such a scale was suggested by Svärm et al. (2017). Recent survey on existing localization methods can be found in Piasco et al. (2018).

One of the key ideas that bypasses the challenge of defining an efficient and robust 2D-3D feature matching required by the abovementioned methods is to use an end-to-end CNN solution that performs both feature extraction and matching. PoseNet Kendall et al. (2015) is an impressive CNN based approach for solving the pose of real images. A dataset of images was used for training Google LeNet Szegedy et al. (2015) where the 6-DoF pose of the camera was used as ground truth. Walch et al. (2017) suggested an improvement to the PoseNet by adding an LSTM, which reduces the dimensionality of the feature vector. Melekhov et al. (2017) used ResNet34 He et al. (2016), which uses encoder-decoder structure to improve model accuracy. Kendall & Cipolla (2016) improved their earlier work Kendall et al. (2015) by applying an uncertainty framework to the CNN pose regressor. In another work, Kendall et al. (2017) studied the affect of various loss functions on the result of PoseNet.

In our study we assume a 3D model of a city is given. Our setup is very challenging since the model and the images consists of only coarse 3D structure of the scene without texture for computing image features. On the other hand, our images are noise-free and there are no object-level occlusions such as trees, cars and people. In addition, we were not limited by data size, as we projected as many images as we chose. Most importantly, our goal differs from that of the aforementioned methods: whereas they focus on obtaining a better and faster solution for geo-localization, we focus on trying to understand the role of geometry, alone, in geo-localization, by systematically training and testing the same neural network on controlled datasets.

## 3 DATA

Using a 3D model as the data source we can sample as many images as necessary in any position, orientation, and resolution. In our study we used a simplified 3D model of Berlin für Wirtschaft und Technologie GmbH (2016). The model contains only the geometry of building walls and rooftops, and does not contain any fine geometric details such as window frames or doors or texture (see bird-eyes view in Fig. 1). We consider three types of images which are projections from this model: edge, face, and depth map, see Fig. 1, and we call them lean images since they contain no texture or structural details. The face images contain the buildings' planar facades.

Each image is defined by its camera pose. That is, $(x, y, \theta, \phi)$, where $(x, y)$ is the position on the ground plane and $(\theta, \phi)$ is the camera $Yaw$ and $Pitch$ angles. We assume for simplicity that the picture is taken at a fixed height, and the roll angle is fixed as horizontal. We generated several image datasets that are sampled uniformly along this 4D grid. The density in the $(x, y)$ domain varies between the data-sets but fixed in the $(\theta, \phi)$ domain. A 6D pose vector (instead of 4D) in the form of $(x, y, q_1, q_2, q_3, q_4)$, is used to represent the camera pose. The $q_1, q_2, q_3, q_4$ are the quaternions representation of the $(\theta, \phi)$ Euler angles. Each set of images is created in a predefined area of the city. The number of images in the set is determined by the size of the area and the grid sampling density. All three types of lean images were generated for each sample pose.

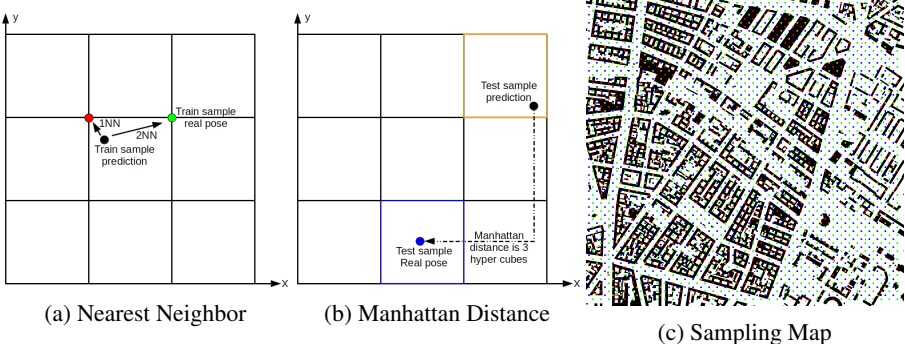

|  |  |  |
|---|---|---|
| (a) Nearest Neighbor | (b) Manhattan Distance | (c) Sampling Map |

Figure 2: (a)-(b) Illustration in 2D of the evaluation measures for geo-matching (a) and geo-interpolation (b). The real measures are 4D in nature. (c) Example of sampling positions on a area of the map. For the training set: green indicates valid samples and red invalid samples. For the test set: blue indicates valid samples and orange invalid samples (please zoon-in on screen).

When dealing with lean images, care must be taken not to include empty images. For example when the camera is facing a building wall from a short distance. Such images contain almost no visual information and do not contribute to the learning process. We discard an image that has less than 8 edges, or an image that does not contain a skyline (at least 50% of its top-most pixel row is sky). Moreover, images of positions inside a building are irrelevant to geo-localization, and are also discarded from training and test (see Fig. 2, right).

## 4  TASKS & HYPOTHESIS

The geo-localization task we consider is to recover the camera pose of an unseen image (not in the training set). We also consider a memorization task of retrieving the camera pose of an image from the training set. For both tasks we consider lean images as an input to the network, and test several configuration of regression and classification networks (see Sec. 5). Our goal is to answer the following questions: (i) Can a CNN be trained to solve these tasks from lean images? (ii) Does geometry play a role when training the CNN for geo-localization?

### 4.1  MEMORIZATION TASK

The memorization task could be defined as a classification task where each $(x, y, \theta, \phi)$ is considered as a class. We examined whether a CNN can solve the memorization task using lean images. Given an image from the **training set**, we tested whether the correct camera pose could be determined. In a sense, the network is trained to overfit, and actually will not generalize. On the other hand, this would mean that the network managed to encode a very large set of images (order of hundreds of the thousand) in some feature space as well as compute an efficient matching function between the features of the images to find the right pose.

**(A) Geometrically Correlated:** In this test, the camera pose for generating the image was used as ground truth for training. Hence, when the pose of nearby images is correlated, the network has access to this geometric information.

**(B) Geometrically Decorrelated:** We aim to test whether the network used the correlation between pose and image, in order to learn the training data. To do so, we randomly shuffled the pose information between images so that poses were not spatially correlated with respect to the images. If no geometry is used by the CNN, the results on this training data are expected to be similar to those obtained with the real pose as a ground truth. Previous study of random labels for object classification task Zhang et al. (2016) also considered randomized labeled for classification, and showed that random labels only affect the time to converge.

**Evaluation:** For a classification network, the evaluation is simply the number of correct classification. For a regression network, the computed pose does not necessarily match exactly a pose of an image from the training set (see Fig. 2a). We used the nearest neighbor (nn) grid sample to the computed pose as the pose retrieval. We report the percentage of images whose correct pose is the

| | Input type | Geo-Matching | | | Geo-Interpolation | | | |
| | | (B) Arbitrary Pose | (A) Correct Pose | | (C) Correct Pose | | | |
| | | | | | 2D $(x,y)$ | | 4D $(x,y,\theta,\phi)$ | |
| | | 1nn | 1nn | 3nn | D<1 | D<3 | D<1 | D<3 |
| Area 400x400 | Edges | 0.45 | 0.97 | 0.99 | 0.64 | 0.82 | 0.58 | 0.75 |
| step 20 | Faces | 0.35 | 0.99 | 0.99 | 0.56 | 0.76 | 0.51 | 0.69 |
| 37K images | Depth | 0.23 | 0.99 | 0.99 | 0.61 | 0.79 | 0.55 | 0.72 |
| | Edges+Faces | 0.29 | 0.98 | 0.99 | 0.72 | 0.88 | 0.65 | 0.82 |
| | Edges+Faces+Depth | 0.24 | 0.98 | 0.99 | 0.71 | 0.88 | 0.64 | 0.81 |
| Area 400x400 | Edges | 0.11 | 0.98 | 0.98 | 0.85 | 0.94 | 0.84 | 0.93 |
| step 10 | Faces | 0.05 | 0.97 | 0.97 | 0.80 | 0.90 | 0.79 | 0.88 |
| 140K images | Depth | 0.06 | 0.97 | 0.97 | 0.83 | 0.92 | 0.82 | 0.91 |
| | Edges+Faces | 0.09 | 0.97 | 0.97 | 0.88 | 0.96 | 0.87 | 0.95 |
| | Edges+Faces+Depth | 0.08 | 0.94 | 0.95 | 0.88 | 0.96 | 0.87 | 0.95 |
| Area 800x800 | Edges | 0.06 | 0.96 | 0.96 | 0.62 | 0.78 | 0.59 | 0.75 |
| step 20 | Faces | 0.01 | 0.96 | 0.96 | 0.51 | 0.68 | 0.48 | 0.65 |
| 170K images | Depth | 0.01 | 0.96 | 0.97 | 0.61 | 0.77 | 0.59 | 0.73 |
| | Edges+Faces | 0.04 | 0.92 | 0.93 | 0.70 | 0.86 | 0.67 | 0.83 |
| | Edges+Faces+Depth | 0.03 | 0.95 | 0.96 | 0.70 | 0.85 | 0.67 | 0.81 |

Table 1: Results of our experiments. The fraction of images on which a correct estimation was obtained out of the total number of valid images evaluated (the higher the better). For geo-matching we use the nearest neighbor measure (nn) and for geo-interpolation the Manhattan distance (D).

nearest neighbor (1nn) and also report the percentage of images whose correct pose is among the three nearest neighbors (3nn) of the computed pose. These evaluations were used for both (A) and (B) tasks. An additional advantage of using this measure is that it is given in grid steps and not in meter/angle units, circumventing the difficulty of comparing distances and angles and enabling a comparison of results with different grid densities (we do provide numerical $\ell_2$ errors in Table 2).

## 4.2 Geo-Localization Task (C)

We tested whether the network can generalize and estimate the pose of an image that is not in the training set. To avoid over-fitting and allow generalization, the network was trained until best result was achieved on a validation set. We do not expect the network to return a correct position that is outside the learned area. For this reason our test set is comprised of images sampled at midpoints of the training grid. These are images that are farthest from the training set samples.

**Evaluation:** In this task we only use regression network. A computed pose is considered correct if it lies within the same grid hyper-cube as the test sample. We report the number of correctly computed poses ($D < 1$). In addition, we considered the Manhattan distance between the hyper-cubes of the computed pose and the test sample (see Fig. 2b). We report the number of images for which this distance is smaller than 3 ($D < 3$). Note that these measurements are also invariant to the sampling step size. Thus, we are able to compare results of experiments that were sampled with different step sizes. For completeness, we also provide the standard $\ell_2$ errors in Table 2.

## 5 Network

For the geo-localization task, a regression network is more natural than a classification network. It allows the network exploiting the geometric structure and information, and to use the same trained CNN for the geo-localization task. It directly returns the pose of the unseen images. Because the considered CNNs were designed for classification tasks, we follow Kendall et al. (2015) and modify them to solve a regression task by simply removing the last softmax layer and replacing it by a fully connected layer of our result vector $(x, y, q_1, q_2, q_3, q_4)$. Although position and orientation are considered as different tasks that should have some weighting factor during the learning process, we noticed that normalizing $(x, y)$ with respect to the total area size eliminates the need for such weighting. Our loss function is $\ell_2$ for the position $(x, y)$ and $\ell_2$ for the orientation $(q_1, q_2, q_3, q_4)$.

We examined several CNN architectures that proved to be successful on object recognition tasks. Specifically, we tested VGG, Google LeNet Szegedy et al. (2015) and ResNet50 He et al. (2016),

| | (A) Geo-Matching | | | | (C) Geo-Interpolation | | | |
|---|---|---|---|---|---|---|---|---|
| | $(x,y)$ | | $(\theta,\phi)$ | | $(x,y)$ | | $(\theta,\phi)$ | |
| | mean | median | mean | median | mean | median | mean | median |
| Area 400x400 step 20 37K images | 3.65 | 3.26 | 0.84 | 0.69 | 26.30 | 11.26 | 10.95 | 1.84 |
| Area 400x400 step 10 140K images | 2.37 | 2.10 | 0.57 | 0.48 | 7.99 | 3.67 | 2.65 | 0.67 |
| Area 800x800 step 20 170K images | 5.43 | 4.71 | 0.67 | 0.54 | 40.23 | 12.28 | 9.80 | 1.40 |

Table 2: Examples of the $\ell_2$ errors for an experiment with Edges+Faces image types in each sub-space: spatial $(x,y)$ errors in (approx.) meters, and orientation $(\theta,\phi)$) errors in degrees. Similar to this example, usually the errors show a long-tail distribution: many images have small errors and a few have very large errors.

built for the ImageNet Large Scale Visual Recognition Challenge (ILSVRC) Russakovsky et al. (2015); Deng et al. (2009). In a set of preliminary experiments we found that ResNet50 had the combination of smallest network size in terms of parameters and the best training and testing results. Therefore, we report our experiments using only the ResNet50 architecture. We trained the CNNs from scratch rather than use the pre-trained weights, since the networks we tested were trained with ImageNet, which contains real photographs. Our assumption is that the pre-trained models are tuned for texture information that is not available in lean images. Transfer learning shows to improve the learning rate, when moving from one AOI to another.

## 6 EXPERIMENTS & RESULTS

We tested and evaluated the regression ResNet50 network for the three tasks described in Sec. 4. The datasets, which are described in Sec. 3, are defined by the following parameters. (**i**) Area of interest (AOI): $(x, y, width, height)$; (**ii**) Grid-step, $\delta$: the distance between adjacent $(x, y)$ position of the sampling grid. That is, adjacent to $(x, y, \theta, \phi)$ are $(x \pm \delta, y, \theta, \phi)$ and $(x, y \pm \delta, \theta, \phi)$. The grid density in $(\theta, \phi)$ domain was fixed; (**iii**) Input type: edges, faces, depth, edges + faces, edges + faces + depth. For the last two input types the images were fed to the network by stacking them channel-wise; (**iv**) Validation set created by randomly choosing 10% of the training samples; and (**v**) Test set created by images that were sampled at midpoints of the training grid.

We used various step sizes for the camera position on the grid: $\delta = 10, 20, 40$ in model units (1 unit $\sim 1$ meter). $\theta$ (*yaw*) was sampled at $5°$ steps between $0°$ and $360°$, and $\phi$ (*pitch*) was sampled at $3°$ steps between $0°$ and $15°$. The height was set to a fixed value of $z \simeq 1.7$ above ground.

The main results of the regression CNN are summarized in Table 1. The results for tasks (A)-(C), each block of three rows consists of a different dataset, defined by the area size and $\delta$. For each block we considered the different types of lean images and evaluated them on the three tasks as described in Sec. 4. Each entry is an average of three different AOIs. For completeness, Table 2 shows an example of the mean and median $\ell_2$ errors of the pose estimation for edges+faces experiment. Similar results were obtained in other experiments. Table 3-4 show the results of testing the limitations of the CNN with respect to the sparsity of the grid ($\delta = 40$) and the size of the datasets ($> 630K$ images). We next discuss the obtained results.

**Memorization Task**
Very poor results were obtained for the memorization task when arbitrary poses were used as ground truth (Table 1–Task (B)), *i.e.* when no geometric correlation between the images and their ground truth was available. The highest percentage of correct matches ($45\%$) was obtained for the smallest set of considered images ($37K$ images). For a largest set ($170K$ images), the percentage of correct matches was less than $10\%$. As can be seen, the quality of the results decreases as the number training samples increases. In contrast, when the correct poses were used as ground truth (Table 1– Task (A)), the CNN succeeded in 1nn localization of more than $92\%$ of the training samples in all cases. We believe the significant differences between the two geo-matching tasks (A) and (B) is due to the network exploiting the geometric correlations when learning a metric between images. This is true, even when we consider only the memorization task, where we evaluate the over fitting of the network.

| | (A) Geo-Matching | | (C) Geo-Interpolation | | | |
| | | | 2D $(x,y)$ | | 4D $(x,y,\theta,\phi)$ | |
| | 1nn | 3nn | D<1 | D<3 | D<1 | D<3 |
| Area 800x800 step 10 636K images | 0.82 | 0.82 | 0.80 | 0.92 | 0.79 | 0.92 |
| Area 1600x1600 step 20 666K images | 0.58 | 0.59 | 0.46 | 0.69 | 0.44 | 0.67 |

Table 3: Testing network learning capacity. These results are from a single experiment where the image input type is only edges. The network ability to learn drops when the number of images grows beyond a certain point.

We also considered much larger datasets with more than $600K$ images (Table 3). The percentage of correct matches dropped to $82\%$ for a dense grid, $\delta = 10$, and to $56\%$ for a sparser grid, $\delta = 20$. For $\delta = 20$ and $> 600K$ images, the network capacity is probably saturated. A comparison of these results to those reported in Table 1 (Task (A)) for the same $\delta$ values, indicates that both the number of images and the grid size determine how successfully the CNN models the data.

In addition, we tested datasets with sparser sampling in the position domain (Table 4 top 2 blocks), and in both the position and the orientation domains (Table 4 bottom 2 blocks). For sparse sampling only in the position domain, the percentage of correct matches is reduced marginally. However, when reducing the sampling also in the orientation domain, the percentage of correct matches is dramatically dropped. This indicates that it is easier for the CNN to model a denser grid (probably because of higher geometric correlation between images), and it is easier to model fewer images (probably because of network capacity).

| | Input type | (A) Geo-Matching | | (C) Geo-Interpolation | | | |
| | | | | 2D $(x,y)$ | | 4D $(x,y,\theta,\phi)$ | |
| | | 1nn | 3nn | D<1 | D<3 | D<1 | D<3 |
| Area 800x800 step 40 61K images | Edges | 0.90 | 0.96 | 0.39 | 0.62 | 0.30 | 0.49 |
| | Edges+Faces | 0.91 | 0.97 | 0.48 | 0.72 | 0.38 | 0.59 |
| | Edges+Faces+Depth | 0.96 | 0.98 | 0.48 | 0.72 | 0.38 | 0.58 |
| Area 1600x1600 step 40 174K images | Edges | 0.89 | 0.90 | 0.22 | 0.39 | 0.19 | 0.32 |
| | Edges+Faces | 0.94 | 0.95 | 0.30 | 0.50 | 0.24 | 0.41 |
| | Edges+Faces+Depth | 0.94 | 0.96 | 0.32 | 0.51 | 0.26 | 0.43 |
| Area 800x800 step 40 / sparse ang. 2.5K images | Edges | 0.40 | 0.41 | Failed | | | |
| | Edges+Faces | 0.37 | 0.38 | | | | |
| | Edges+Faces+Depth | 0.26 | 0.27 | | | | |
| Area 1600x1600 step 40 / sparse ang. 7K images | Edges | 0.16 | 0.18 | Failed | | | |
| | Edges+Faces | 0.17 | 0.19 | | | | |
| | Edges+Faces+Depth | 0.13 | 0.14 | | | | |

Table 4: Low grid density results. Datasets (single experiment each) with sparser spatial sampling (top two blocks), and sparser spatial and orientation sampling (bottom two blocks) where the pitch angle is $\in [6, 12]$, and yaw $\in \{45i\}_{i=0}^{7}$. The sparser the data, the worse the results. Geo-interpolation could not succeed in very sparse and very small datasets.

**Geo-localization task**
Here we tested whether the pose estimation by the CNN generalizes to unseen images. We used the same training as in the memorization task with the correct pose as a ground truth, and we tested it on images sampled from the mid point of each grid cell. We report our results with respect to the 2D position as well as with respect to the 4D parameters of a pose (Table 1).

The network was able to generalize image position with good accuracy where $\sim 70\%$ of images are correctly positioned in their grid cell, and above $80\%$ of the computed poses are within three cells of the correct one. As expected, this task achieves better results on a tighter grid ($\delta = 10, \sim 88\%$) than on a sparse grid ($\delta = 20, \sim 70\%$). The 4D position error is lower bounded by the 2D position error, and hence is greater. Moreover, the sampling rate in the orientation domain is much higher that in the location. Hence a small error in orientation estimation has a greater effect on the 4D errors. Still, the accuracy in 4D for $\delta = 10$ is $\sim 87\%$.

It is clear from Table 1–Task(C) that for $\delta = 10$ the results are better than for $\delta = 20$, even if the number of images is larger. We further explore the effect of the grid density for a sparser grid, $\delta = 40$, where the percentage of correct estimations dropped significantly below 50% and 30% for $61K$ and $174K$ images, respectively (Table 4-Task(C)). For $\delta = 10$ for $636K$ images, 80% of the estimations were correct (Table 3-Task(C). For this task, sparser sampling is more critical than the memorization task as can be seen in Table 4. For very sparse sampling of the $4D$ space the network cannot really generalize to positions not seen before. Here again we believe that not only the number of images play a role but also their density. The denser the grid, the higher the correlation between images, and hence better generalization can be obtained.

A nice application of our results is the ability to rate the distinctiveness of positions in the city. In Fig. 1 we illustrate how certain places can be easily recognized (high geo-interploation success rate) while other are more difficult. Note, for instance, how open spaces are more distinct than narrow streets.

**Effect of Data Type**
Faces alone provides the least geometric information, and indeed in most cases was inferior to edges or depth results. Surprisingly, edges alone provides better information than depth alone for all tasks. Similar results were obtained for all data types for the memorization task for $\delta \leq 20$ (Table 1), but for a very sparse grid $\delta = 40$ (Table 4), richer geometric information improves the results. We believe it is because the results on $\delta \leq 20$ were very high to begin with only edges.

For the geo-localization task (C), adding the faces information significantly improved the results, as expected. Surprisingly, the depth information did not show any significant performance gain when $\delta \leq 20$. This may indicate that edges+faces provide sufficient information for these cases. However, for a very sparse grid, $\delta = 40$, with a relatively small number of images, adding the depth significantly improves the results (Table 4, 174K images).

For the data with geometrically decorrelated pose (Task B) and for the very sparse sampling (Table 4 bottom 2 blocks), the more information we add, the worse results were. The reason for this is still unclear to us. A possible explanation is that as the problem becomes more of a memorization task, the increase of information makes it harder for the CNN to find discriminant features.

**Classification**
For completeness, we also considered a brute-force classifier networks. Using the number of images in the training set as an output ($\sim 100k$ images), creates a huge FC last layer, which results in a much larger number of parameters in comparison to the regression network. Hence we test a network with a binary representation of 100k indexes (17 output parameters). Experiments showed that the classification CNN is able to memorize the training set extremely well as also shown by Zhang et al. (2016). On all experiments, on both step 10/20, the result exceeded 97% accuracy. On the other hand, as expected, this network gave very poor results for the geo-loclization task even for $\delta = 10$ (below 15% and 21% on $D < 1$ and $D < 3$, respectively) compared to the regression network (above 85%). The test sample location estimation is done by feeding the CNN with a test image and than taking the location of the training sample that matches the model output. We believe this proves that such classification supervision is inadequate for the Geo-Localization task, the network has no incentive for creating any meaningful features that would later be applied to new similar images.

# 7    CONCLUSIONS

In this work we showed that (i) CNN can achieve good results in geo-localization tasks using only lean images taken from a very simple 3D model, and (ii) that geometry plays an important role in geo-localization, by achieving good results while ignoring texture and scene details. The results indicates that noise-free lean images are sufficient for solving the memorization task using a CNN, and that the use of uncorrelated images makes it nearly impossible. In addition, our results indicate that (iii) the generalization task, can also be solved by CNNs when using lean images.

From a more practical perspective, it is of interest to explore whether geometric information can be used for real life geo-localization tasks, also because 3D models, *e.g.*, the Open Street Map project Wiki (2018) are readily available.

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
