# OpenReview forum: "Lean Images for Geo-Localization"
_ICLR.cc/2020/Conference — Reject_

### Official Review · AnonReviewer1 · 2019-10-19
**Official Blind Review #1**

**Rating:** 3

**Review:**

This paper evaluates the performances of Deep Learning for geo-localization tasks in a world of images without textures ("lean images"). More exactly, the lean images are images rendered from a 3D model of a city. made of the depth and/or the buildings' edges and/or the buildings' faces. For the purpose of the evaluation, no real images are used,  only lean images both at training and test times. Many lean images are generated for training a network to predict the camera pose (2 translation parameters,  two angles), as a classification problem or a regression problem.

The motivation for the study is to understand the use of geometric aspects by a deep network to solve the task. However, as the paper presents mostly the results of the quantitative evaluation, it is not clear what we really learn about the mechanisms of the deep network. I think the empirical results can be interesting for people working on geo-localization, but probably not for the audience of ICLR.

I suspect the authors have in mind for the future to learn to predict a lean image from a real image. I would find this aspect very interesting. It would also be interesting to evaluate the performance on the  geo-localization task when noisy images are used as input.


**Experience Assessment:**

I have published one or two papers in this area.

**Review Assessment: Checking Correctness Of Derivations And Theory:**

I carefully checked the derivations and theory.

**Review Assessment: Checking Correctness Of Experiments:**

I carefully checked the experiments.

**Review Assessment: Thoroughness In Paper Reading:**

I read the paper at least twice and used my best judgement in assessing the paper.

---

### Official Review · AnonReviewer2 · 2019-10-24
**Official Blind Review #2**

**Rating:** 3

**Review:**

The paper tackles the problem of geolocating an image from a 3D model of a city. This is an interesting and challenging problem since it involves solving the matching under 3D transformations at large scale.

The proposed approach is to use geometric features such as edges, faces and depths of the image, which is called "lean image" by the paper. The paper demonstrate that by using the "lean image" a neural network can recognize its geolocation with substantial accuracy. However, the idea of using geometric features has been investigated in quite a few previous literature and the paper fails to discuss relevant works, e.g., [1,2,3] and distinguish their differences.
[1] Li et al. Planar Structure Matching Under Projective Uncertainty for Geolocation. ECCV 2014.
[2] Mousavian and Kosecka. Semantic Image Based Geolocation Given a Map. ArXiv 2016.
[3] Kim et al. Learned Contextual Feature Reweighting for Image Geo-Localization. CVPR 2018.

The paper contains mostly empirical evaluations however the provided experiments do not well support the claim that CNN works well with geometric structures in geolocalization. The paper should compare with CNN on RGB images on the same data in order to demonstrate its effectiveness.

Review:
## Contribution of the paper:
1. The paper argues that "lean images" (geometric structure of a 2D scene) contains sufficient information for solving the memorization of geolocation tasks.
2. The paper provide an approach to evaluate the argument.
3. The paper provide experimental validation that the proposed method works fairly on a certain Berlin dataset.

## Feedback
The paper is interesting but it does not meet the standard of this conference. There are some key reasons:
1. The writing is poor. The paper is hard to follow. There are also missing references. The importance of geometry in geolocation has been studied in many previous works.
2. The paper is empirical so it has to provide strong empirical evidence to support the claim. The experiment is only conducted in Berlin data whose scope is fairly small. Without large scale experiments, it is hard to convince readers that CNN is able to "memorize" all the locations using just geometric features. Without comparing to RGB baseline or other existing approaches, it is hard to convince readers that the proposed approach works fairly well. It also seems the input images are just 2D projections so without experiments using real 2D images, it is hard to say the approach is effective.

## Improvements that could be done
1. Please make sure everything in the tables are clearly explained. The metrics should be explained somewhere in the caption. It is hard to find the meaning of some notations in the table such as "37K images", "Area 400x400". Please clearly state the unit.
2. Please include some baseline results into the tables so that readers know how much improvments achieved by your approach.
3. Please reorganize the paper to make it more clear.
4. Please provide more literature review and discuss the difference of your work to existing ones.


## Questions
1. How large is the Berlin data?
2. How would your method work when there is only a street image? How would you method scale when 3D model is not always available.
3. How would you method work when there is some places without much structure? Rural area could be an example.

**Experience Assessment:**

I have published in this field for several years.

**Review Assessment: Checking Correctness Of Derivations And Theory:**

I assessed the sensibility of the derivations and theory.

**Review Assessment: Checking Correctness Of Experiments:**

I assessed the sensibility of the experiments.

**Review Assessment: Thoroughness In Paper Reading:**

I read the paper at least twice and used my best judgement in assessing the paper.

---

### Official Review · AnonReviewer4 · 2019-11-04
**Official Blind Review #4**

**Rating:** 3

**Review:**

The paper evaluates a geo-localization task based on "lean" images only, obtained by projection of 3d models without texture information. Multiple levels of granularity of the lean images (edges/edges+faces/edges+faces+depth) are compared for the learning, both in a "memorization" setting and in a "generalization to unseen poses" setting. Moreover, the behavior of the learning when labels are shuffled is evaluated.

The paper is motivated by identifying lean images with "pure geometrical" representation, and argues that the study is a demonstration that the network can learn some spatial map. I do not share this analysis: the edges, shapes and colors of the lean images still create some characteristic shapes that do not have to be linked with the 3D map in order to be detected; it is not clear why the detection from lean images is more geometrical in essence w.r.t. textured images. The motivation of the paper is therefore not compelling, and the study would have more relevance as an ablation study of a method using both geometry and texture. Can you give more compelling arguments for the problem setup, rather than a mere thought experiment?

Other than this particular setup of geo-localizing based on lean images only, the novelty of the paper and the approach seem limited and does not offer new insights in geo-interpolation methods, geometry-aware neural networks, or memorization. Therefore, I believe the paper does not meet the standards of ICLR.


**Experience Assessment:**

I have read many papers in this area.

**Review Assessment: Checking Correctness Of Derivations And Theory:**

I carefully checked the derivations and theory.

**Review Assessment: Checking Correctness Of Experiments:**

I carefully checked the experiments.

**Review Assessment: Thoroughness In Paper Reading:**

I read the paper at least twice and used my best judgement in assessing the paper.

---

### Decision · Program_Chairs · 2019-12-19

**Decision:**

Reject

**Comment:**

The submission studies the problem of geolocalizing a city based on geometric information encoded in so called "lean" images.  The reviewers were unanimous in their opinion that the submission does not meet the threshold for publication at ICLR.  Concerns included quality of writing, novelty with respect to existing literature (in particular see Review #2), and limited validation on one geographic area.  No rebuttal was provided.